# Relationship between Body Chemical Composition and Reproductive Traits in Rabbit Does

**DOI:** 10.3390/ani11082299

**Published:** 2021-08-04

**Authors:** Meriem Taghouti, Javier García, Miguel A. Ibáñez, Raúl E. Macchiavelli, Nuria Nicodemus

**Affiliations:** 1FeedInov CoLab, Integrated Production Systems, Quinta da Fonte Boa, 2005-048 Santarém, Portugal; myriam.taghouti@feedinov.com; 2Departamento de Producción Agraria, Escuela Técnica Superior de Ingeniería Agronómica, Agroalimentaria y de Biosistemas, Universidad Politécnica de Madrid, C/Senda del Rey 18, 28040 Madrid, Spain; javier.garcia@upm.es; 3Departamento de Economía Agraria, Estadística y Gestión de Empresas, Escuela Técnica Superior de Ingeniería Agronómica, Agroalimentaria y de Biosistemas, Universidad Politécnica de Madrid, C/Senda del Rey 18, 28040 Madrid, Spain; miguel.ibanez@upm.es; 4Colegio de Ciencias Agrícolas, Universidad de Puerto Rico, Mayagüez 00681-9000, Puerto Rico; raul.macchiavelli@upr.edu

**Keywords:** body composition, fertility, kits born alive, rabbit does

## Abstract

**Simple Summary:**

At the beginning of the productive life of rabbit does, there must be a balance between ensuring at least a minimal degree of bodily development to guarantee a successful reproductive life, and the minimization of the unproductive rearing period, but nowadays there is no clear recommendation about the optimal moment for the first artificial insemination (AI). A better body condition at the first AI (higher body protein, fat and energy), that indicates a higher degree of maturity of the rabbit doe, did not influence fertility at the first AI (that is usually very high), but improved it at the second AI (that is usually lower than the first one). The percentage of kits born alive at the first and at the second AI also were positively influenced by the body protein content at the first AI. We can conclude that the degree of maturity at the first AI is a key point to optimize the does reproductive success, with body fat and body protein content being relevant factors.

**Abstract:**

The relationship among live weight, chemical body composition and energy content (at artificial insemination (AI) and three days before parturition), estimated by bioelectrical impedance with fertility rates and the percentage of kits born alive, was studied during the first three AI. The first AI was conducted at 16 weeks of age in 137 rabbit does that weighted 3.91 ± 0.46 kg. Their body chemical composition was 17.4 ± 0.50%, 16.1 ± 2.6%, 1067 ± 219 kJ/100 g body weight, for protein, fat and energy, respectively. An increase in body protein, fat and energy content at the first AI did not affect fertility at the first AI but improved it at the second AI (*p* ≤ 0.030). Moreover, an increase in body fat and energy content at the second AI improved fertility at the second AI (*p* ≤ 0.001). Fertility at the third AI was positively influenced by body protein at the third AI and the increase in body protein and fat between the second parturition and the third AI (*p* ≤ 0.030). The percentage of kits born alive at the first and at the second AI improved with the increase in body protein at the first AI (*p* ≤ 0.040). In conclusion, a minimal body protein and fat content is required at the first AI to optimize the reproductive performance in young does.

## 1. Introduction

In the last four decades, rabbit production underwent a noticeable change from a traditional and familiar organization to industrial and intensive systems. Consequently, genetic selection programs and new breeding management systems were established, improving the production of the new hybrid lines used. This has led to an increase in female nutritional requirements [1,2], health problems [3,4,5,6,7] and welfare necessity [8,9,10,11]. Nutritional strategies in reproductive females need to be global and consider both the short-term productive factors (litter size, milk production, or fertility) as well as the long-term factors (body condition or health). Therefore, the expected improvement in nutritional management should be based on an accurate analysis of the requirements of the doe, its evolution during successive reproductive cycles and the identification of crucial moments in the life of rabbit does to optimize productivity and longevity.

Two of the key points of the reproductive success of rabbit does are their birth weight and maturity at the first artificial insemination (AI). There is an optimal threshold for birth weight (>57 g) to optimize the initial reproductive performance, which is associated with an increase in the live weight and fat reserves at the first AI [12,13,14]. The latter are the traits used to define the maturity of rabbit does at the first AI, and both are also related to the nutritional rearing strategy and the time of AI [13,15]. However, there is no clear recommendation indicating the weight and/or body condition of the nulliparous rabbit doe at the onset of its reproductive life [16]. In the current management systems, does are inseminated at a fixed age with minor or no consideration of their weight and chemical body composition. As in other species, there must exist a balance between ensuring at least the minimal degree of body development needed to guarantee a successful reproductive life, and minimizing the unproductive rearing period. Accordingly, the study of the body chemical composition of rabbit does seems to be a useful tool not only to improve feeding, but also for general rabbit doe management [17]. The final aim is to extend the lifespan of rabbit does, which is limited by the relatively high early mortality and culling rate in intensive production systems [5].

A new non-destructive technique to estimate rabbit doe body chemical composition (moisture, ash, protein, fat, and energy content) based on the bioelectrical impedance measurement, live weight and physiological status of rabbit does, was developed by Pereda [18]. This method is easier and cheaper than TOBEC [19], and in both methods, the variations of gut contents are included in the error term. Furthermore, it allows the prediction of total fat and energy content (not only the perirenal fat content, as it is with the case with the ultra-sound technique [20]), as well as the body protein, which is not usually estimated with other methodologies to evaluate body condition.

The aim of this work was to establish the relationship between chemical body composition (at AI and parturition), determined using bioelectrical impedance, and both fertility and the percentage of kits born alive during the first three inseminations, and to identify the most important moments to record the body chemical composition.

## 2. Materials and Methods

### 2.1. Animal Husbandry and Management

Data of the estimated body chemical composition, the fertility and percentage of kits born alive recorded in two different farms in 2010 were used in order to obtain a wide variation in chemical body composition. In farm A, 106 crossbred (New Zealand White × Californian) rabbit does from the UPV hybrid line (genetic line selected for prolificacy in Polytechnic University of Valencia in Spain using the crossline A × Line V) were used. After the first parturition, does were submitted to three different breeding systems, defined by the parturition–AI and parturition–weaning intervals (4/32, 11/35 and 14/42, Table 1) to obtain a wide variability of body condition. In farm B, 37 crossbred (New Zealand White × Californian) rabbit does from the Hyplus hybrid line (prolific hybrid maternal line selected by Hypharm in France) were used and they were inseminated 11 d after parturition and litter weaned at 35 d of age. Rabbit does were inseminated for the first time between 16 and 18 weeks of age. Rearing period of rabbit does was not controlled and they were fed ad libitum after the first insemination. Non pregnant-non lactating rabbit does were restricted to around 150 g/d. Two commercial diets were used, one in farm A containing 17.5% crude protein, 32.0% neutral detergent fibre and 9.95 MJ digestible energy/kg (based on dehydrated alfalfa, wheat bran and barley), and another in farm B containing 16.5% crude protein, 30% neutral detergent fibre, and 10.8 MJ digestible energy/kg (based on dehydrated alfalfa, sugar beet pulp, sunflower meal, rye and wheat). All rabbit does were submitted to a cycle of 16 h light/8 h dark, and heating, cooling and forced ventilation systems allowed the building’s temperature to be maintained between 18 and 24 °C. Animals were handled according to the principles for the care of animals in experimentation published by the Spanish Royal Decree 53/2013 [21] and favourably assessed retrospectively by the Ethics Committee of the Polytechnic University of Madrid.

The relationship between body chemical composition and fertility was studied during the first three cycles as the number of replicates decreased from the third parturition onwards (Table 2). The percentage of kits born alive was only studied at the first and second parturition (N = 121 and 83, respectively) due to the reduction in replicates in the third parturition. The range of variation in chemical body composition of these does at the moments of insemination and parturition is shown in Table 2.

Seminal doses with more than 20 million spermatozoa in 0.5 mL of a commercial diluent (Magapor S.L.) were made using a pool of fresh heterospermic semen from bucks selected for growth performance. In order to synchronize the oestrus in the second, third and fourth insemination, 48 h before insemination, the does were injected with 25 IU of eCG (Equine Chorionic Gonadotropin, Segiran, Lab. Ovejero, León) [22]. On the day of insemination, the does received an intramuscular injection of 10 µg of buserelin Suprefact^®^ (Hoechst Marison Roussel, S.A., Madrid). Buserelin is a Gonadotropin-releasing hormone agonist (GnRH agonist), which is used to induce ovulation in rabbit does [23].

### 2.2. Experimental Procedures and Data Recording

In order to determine the chemical body composition of the does, a bioelectrical impedance analysis (BIA) technique was used [18,24]. Measurements were taken with the four-terminal analyzer Quantum II (Model BIA-101, RJL Systems, Detroit, MI, USA). This device generates an alternating current of 425 µA of intensity at a frequency of 50 kHz. It is provided with 2 black electrodes to conduct the electrical current through the doe’s body, and 2 red electrodes to register the resistance and reactance resulting from the passing current. When it is used with rabbit does, a needle (Terumo, 21G × 1 ½′, 0.8 × 40 mm, nr 2) is inserted in the extremity of each electrode. The needles must pass through the skin of the rabbit doe at four reference points along the loin (two near the scapula and other in the distal part of the loin). Impedance is defined by the equation: Impedance = (Resistance^2^ + Reactance^2^)^1/2^. These three parameters, in addition to the physiological status, live body weight and parturition order were used further to estimate the body composition of the rabbit does. Rabbit does were weighed and their body composition estimated on the days of artificial insemination and parturition (three days before parturition, both pregnant and non-pregnant does) during the first three reproductive cycles. The regression equations developed and validated by Nicodemus et al. [24] and Pereda [18] enabled the prediction of the doe’s body content in moisture, protein, fat and ash expressed as percentages or as g/100 g of body weight. Furthermore, energetic body content expressed in MJ or in kJ/100 g of body weight could be also estimated.

### 2.3. Data Treatment and Statistical Analysis

In order to establish correlations among parameters of chemical body composition (moisture, fat, protein, ash and energy) at the moments of insemination and parturition during the first three parities, we used Pearson’s correlation coefficient (CORR procedure of the SAS system) (SAS Inst., Cary, NC, USA). The GENMOD procedure of the SAS system [25] was used to study the relationship between chemical body composition, fertility during the first three parturitions and the percentage of kits born alive in the two first parturitions, as the logistic regression [26,27] is an adequate tool for modeling proportions [28]. In the generalized linear models used, the link function employed was ‘logit’ (Equation (1)) and we considered that both fertility and the percentage of kits born alive were proportions arising from a binomial distribution. In equation 1, ‘p’ is the mean of the proportion of fertility and kits born alive.
Logit (p) = ln (p/1 − p); p ∈ (0,1)(1)

Our first aim was to determine whether the variations in fertility and percentage of kits born alive (dependent variables) were significantly affected by the body chemical composition and its variation between insemination and parturitions (independent covariates). The model also included, as fixed effects, the farm and the breeding system (except for nulliparous does). Afterwards, only significant covariates were retained for the interpretation of coefficient estimates (β) and the linear predictor (η = x’β). In order to display the covariates’ effects, three levels (low, medium and high), representative of the range of variation of each covariate, were fixed for each significant covariate (body constituent) based on the data used in this investigation. To predict the means of fertility and percentage of kits born alive expected for each level (the mean or predicted values, ‘p’) the inverse logit function was used (Equation (2)).
p = e^η/^(1 + e^η^)(2)

Among the criteria of the goodness of fit, we used the likelihood ratio chi-square, ‘Q_L_’, also called ‘deviance’, to establish whether models fitted appropriately [29]. The inverse function of the logit transformation was also used to calculate a confidence interval for the expected percentages of fertility and kits born alive for each level [29]. In fact, in the SAS output and for each estimator calculated, there are the values of lower (Lη) and upper (Uη) limits for its 95% confidence interval. In this case, the method consists in transforming the linear values of the limits to the log-link scale using Equation (3).
Upper limit p = e^Uη^/(1 + e^Uη^); Lower limit p = e^Lη^/(1 + e^Lη^)(3)

To calculate a standard error for the mean of predicted values, ‘p’, for each level of the covariates retained, we used the Delta method [30], which involves the standard error of the coefficient estimate (SEη) as well as the predicted value, as explained in Equation (4).
SEp ≈ p(1 − p) SEη(4)

The results were transformed from the logit scale.

## 3. Results

### 3.1. Correlation among Does’ Corporal Constituents during the First Three Parturitions

The initial live weight and chemical composition of rabbit does had a wide range of variation, which was more important for the fat than for the protein content (16 vs. 3%, respectively, for coefficient of variation (Table 2)). Rabbit does showed an important degree of growth between the first and the second AI (3906 vs. 4187 g, weight of does at 1st and 2nd AI, *p* < 0.05 (Table 2)). Live body weight was positively correlated (*p* < 0.001) with fat (r = 0.43 to 0.83) and energy body content (r = 0.43 to 0.71), and negatively with moisture (r = −0.35 to −0.72) (expressed on % and MJ/100 g body weight, respectively) from the first AI to the third parturition (Table 3). In this period, body fat was inversely and closely correlated with moisture (r = −0.95 to −0.99), and positively with body energy (r = 0.79 to 0.98), while body protein was positively correlated with ash content (r = 0.30 to 0.83). Regarding moisture, a negative correlation was observed with energy (r = −0.43 to −0.71).

Protein content at the first AI was not correlated with protein content at any other moment of the cycle (Table 4). However, protein content was positively correlated at parturition and the subsequent AI (both at the first and second parturition; *p* < 0.001). In contrast, the body fat content was mainly positively correlated in the period between the first AI and the third parturition, and the same was observed for body energy content (*p* < 0.05 (Table 4)).

### 3.2. Relationship between Body Condition and Fertility during the First Three Parturitions: Nulliparous, Primiparous and Multiparous Rabbit Does

The fertility rate registered in the first parturition was 93.4%. There were no differences between data from the two farms used for the analysis (*p* > 0.05). Neither live body weight, nor its chemical composition at the first AI, affected the fertility rate at this moment. In fact, there were no differences in the initial body composition and live weight at the first AI between pregnant and non-pregnant does (Table 5), although these results have to be taken with caution due to the small number of non-pregnant rabbits.

As expected, in the second parturition, fertility was lower than in the first parturition (56.2 vs. 93.4%, respectively; *p* < 0.05). Breeding system and farm did not have any effect on fertility in the second parturition. Recorded fertility means were 54.0%, 50.8% and 65.8% for the studied breeding systems R4, R11, and R14, respectively. Live body weight at the moments of first AI, first parturition, and second AI did not affect the fertility rate in the second parturition. Fertility in the second AI was related to chemical body composition at the first AI, in contrast to that observed at the first parturition. Body protein content (*p* = 0.007), fat (*p* = 0.030), and energy (*p* < 0.001) at the first AI positively affected the fertility rate in the second AI (Table 6). The relationship between these parameters and fertility is lineal in the logit scale. Consequently, the model was fitted using three fixed levels of body protein, fat, energy, and protein/energy ratio. These levels were chosen to cover the data range used for the analysis. Fat and energy were positively correlated at the first AI (r = 0.79, *p* < 0.001), and with their values at the second AI (r = 0.33 and 0.74; *p* < 0.001 (Table 3 and Table 4)), while at the first AI protein was negatively correlated with fat but did not have any correlation with energy, and was not related to protein at the second AI. The three levels determined for each constituent and used in the model are shown in Table 6. The increase in body protein from 16 to 18% and body fat from 10 to 20% increased fertility from 19.9 to 72.0% and from 31.3 to 69.5%, respectively. The results showed that does with higher energy content (1400 vs. 700 kJ/100 g) underwent improvements in fertility from 19.1 to 83.6%.

A higher body protein/energy ratio at the moment of the first AI negatively influenced the fertility rate in the second AI (*p* < 0.001 (Table 6)). This ratio reflects again the importance of the energetic content, since high ratios (30 g/MJ) corresponded to low corporal energy (800 kJ/100 g) rather than to high body protein content. In fact, the latter showed lower variability (2.9 vs. 20.5%, coefficient of variation of body protein and energy at the first AI, respectively (Table 2)).

Chemical body composition at the second AI was also related to fertility in the second AI, as detailed in Table 7. Fat and energy contents at the second AI were positively related to fertility at this moment (*p* < 0.001) in a similar way to their relation at the first AI. These two variables were positively correlated both at the first and at the second AI (*p* < 0.001 (Table 3)), and were also positively correlated with fat and energy content at the first AI (*p* < 0.001; Table 4), although their coefficient of variation increased at the second AI (Table 2).

Body protein at the second AI did not exert any effect on fertility at the second AI, in contrast with its observed effect at the first AI. It was positively correlated with fat and energy content at the second AI. Protein contents at the first and at the second AI were not correlated (Table 4), although their average values and variability were similar (Table 2), and were lower than body protein values observed from second parturition onwards. A higher body protein/energy ratio at the second AI again negatively affected the fertility rate at the second AI, reflecting the effect of energy content at the second AI.

Moreover, to study the factors that influenced the fertility at the second AI, multiple linear regressions were fitted considering two independent variables: live weight or body composition at the first AI combined with their respective variations between the first and second AI (Table 8). Live body weight at the first AI and weight gain during the interval between the first AI and the first parturition as well as during the interval between the first and second AI were positively related to fertility at the second parturition.

The fertility rate recorded at the third AI was 73.9%. No effect of breeding system or farm was found. Recorded fertility means were 56.2.0%, 90.3% and 75.8% for the studied breeding systems R4, R11, and R14, respectively, showing differences in the first two values (*p* = 0.007). The fertility rate recorded at the third AI was not affected by body composition (fat and energy) at the moments of the first and second AI. Unlike the fertility at the first and second AI, we registered a positive effect of protein content on fertility at the third AI (*p* = 0.030 (Table 9)). Body protein content at the third AI and second parturition were positively correlated (Table 4), and none of them were correlated with body fat or energy content (Table 3). Moreover, we studied the effect of the changes in chemical components between consecutive AI and parturitions. A positive effect of body protein gain between the first and third AI (*p* = 0.020), and between the second parturition and third AI (*p* < 0.001 (Table 9)), was observed. Similarly, body fat gain between the second parturition and third AI was also positively related to fertility rate at the third AI (*p* = 0.020 (Table 9)).

The effect of the previous reproductive success (or not) is reflected in the strong effects of both protein and fat gain before the third AI on the fertility at the third AI. In fact, pregnant does at the third AI showed lower fertility at the second AI, which allowed them a better body reserve recovery (especially protein) at the third AI compared to non-pregnant does (that showed higher fertility in the previous AI (Table 10)). This is confirmed by the positive relationship between fertility rates at the third and first AI (*p* = 0.050), and the negative relationship between fertility at the third and second AI (*p* = 0.050). We observed that does that gave birth at the second parturition had lower chances of becoming pregnant at the third AI (*p* = 0.043; Table 10).

### 3.3. Relationship between Body Condition and Percentage of Kits Born Alive during the First Two Parturitions: Nulliparous and Primiparous Does

The percentage of kits born alive out of the total born in the first parturition was 93.4%, and the number of kits born alive per doe was 8.00 ± 2.97. Among chemical body constituents, body protein (*p* = 0.040) and energy contents (*p* = 0.010) at the first AI increased the percentage of kits born alive (Table 11). Both variables were not correlated at the first AI (Table 3). The relationship between these parameters and the percentage of kits born alive is linear in the logit scale. Consequently, the model with logit link was made using three fixed levels of body protein and energy contents. These levels were chosen to cover the data range used for the analysis. When protein content at the first AI increased from 16 to 18%, the percentage of kits born alive increased from 88.7% to 95.2%, respectively. Furthermore, when body energy at the first AI increased from 900 to 1300 kJ/100 g, the percentage of kits born alive increased from 92.4 to 95.0, respectively.

Body fat content was negatively correlated with body protein and positively correlated with energy content at the first AI (r = −0.21 and 0.79, respectively; *p* < 0.05 (Table 3)), and no effect on the percentage of kits born alive was observed. Live weight at the first AI was also correlated with body protein, energy and fat contents (r = −0.46, 0.65 and 0.83, respectively; *p* < 0.001 (Table 3)) but was not related to the percentage of kits born alive. Furthermore, a higher percentage of kits born alive was observed in farm B compared to farm A (98.1 and 88.7%, respectively; *p* < 0.001), but the number of kits born alive per doe was not different between the two farms.

The percentage of kits born alive recorded in the second parturition was 87.5% and the number of kits born alive per doe was 10.3 ± 4.46. This rate is lower than that observed in the first parturition (93.4%), but the number of kits born alive increased (8.00 ± 2.96 kits born alive/doe in the first parturition). No effect of the breeding system or farm on the percentage of kits born alive at the second parturition was reported. Weight and body composition at the first AI were not related to this trait. However, a higher body protein at the second AI (*p* < 0.001) was associated with an increase in the percentage of kits born at the second parturition (Table 12).

## 4. Discussion

A general problem observed in rabbit does is their low fertility rate in the second parturition [31]. It is usually explained by the energy deficiency observed before the second insemination [16]. In this period, it might be difficult for the rabbit doe to meet the requirements for both pregnancy and growth due to the limited feed intake [1,32,33,34,35]. However, when the energy supply was increased, it was dedicated to milk production with no limitation of reserve mobilization [36,37]. In this way, Pascual [13] hypothesized that this negative energetic balance would be a natural adaptation to optimize evolutionary success. In this context, it is interesting to study the relationship between the factors related to body chemical composition and their influence on fertility and kit survival at birth, in order to identify the threshold to be met by rabbit does at the beginning of their reproductive life.

The evolution of the live weight and chemical composition of rabbit does from their first AI onwards indicated that they were still growing when inseminated the first two times, which agreed with recent data [38]. Live body weight, body energy and fat were closely and positively correlated, which was similar to the correlation between perirenal fat thickness and body energy content reported previously [39,40]. In contrast, body protein had a minor or no correlation with the latter traits, but a negative one with live weight from the second parturition onwards. These results agree with rabbit doe maturation in this period, which would depend on maturity at the first AI and on reproductive success. Once maturity is reached (or nearly reached), the changes in body weight might be mainly associated with fat mobilization and/or deposition. This would agree with the moderated and positive correlation between live weight and body condition score [41].

The differences in chemical body composition and live weight between the first AI and the first parturition seemed to be related to the success in the first insemination, which was not influenced by body condition or live weight. This effect was reported by other authors [2,34,42]. It may be explained by the specific situation of non-pregnant does, which would use the entire intake for body protein and fat accretion, and accordingly, energy accretion, towards the completion of the final step to reach their maturity, where the fat deposition is much higher than the protein deposition (24 vs. 6% increment, respectively). Meanwhile, pregnant does have to supply gestation requirements that are especially important during the last 10 days of gestation, and which can impair not only fat content [17,43,44], but also protein balance [45], with respect to non-pregnant does. It must be taken into account that rabbit does inseminated later show a higher feed intake capacity [46] and lower growth requirements. Furthermore, rabbit does reduce feed intake in the days before parturition, contributing to the impairment of their nutrient balance [43].

The impairment of fertility in the second insemination reflects the specific situation of primiparous rabbit does, which suffer a negative energetic balance during their first pregnancy and lactation, compared to non-pregnant does, which seems to negatively influence reproductive performance and especially fertility [2,32,34], although this could be considered a natural adaptation, as commented before [13]. Live body weight at the moments of first AI, first parturition and second AI did not affect the fertility rate in the second parturition. This agrees with the findings of Rommers et al. [46], who did not observe any effect of body weight at the first AI on fertility rate in the first two parturitions. Nevertheless, the combination of a high live weight and high weight gain between the first two AI was also related to better fertility. Rabbit does that lose weight between the first two AI, regardless their initial body weight, were unable to present an acceptable reproductive performance in the second cycle. In this period, live weight was positively correlated with body fat and energy content, but not with protein, as does are finishing their protein accretion. The relationship between the fertility in the second AI and the chemical body composition at the first AI confirms the importance of the rearing management of rabbit does. Diets used [2,14,42,47] and the time of first insemination [15,48,49,50] influenced body composition, which consequently affected fertility. Therefore, at the end of the rearing period, reproductive does should reach an optimal body condition (minimal body protein, fat and energy content), assuring an adequate feed intake and body development, which enable high fertility rates during first parturitions [16]. In this sense, Pascual et al. [13] stated that body data at the first AI are a sign of doe soma and might be related to its productive potential. In this context, the supplementation of reproductive sows with certain daily amounts of amino acids enabled an adequate retention of nitrogen that led to an acceptable reproductive performance [51]. In this study, the threshold in the body composition at the first AI to avoid a sharp reduction in fertility in the second insemination might be set at 18% protein and 20% fat, but few does met this condition (12 and 7%, respectively). Other studies where the time of first insemination was delayed (to 18.4 or 19.5 weeks of age) rendered an increase in the body protein at that moment, as well as higher fertility values (83–87%) that were also associated with a higher fat content [38], although this was not always the case [52]. When the latter two studies were considered together, the does that were successful in the first five AI were lighter and had less body fat than the average (although their mean and range were similar to the current study), and the same body protein (although it was in the upper threshold that was previously mentioned: 17.9%) [53], suggesting the potential relevance of body protein at the beginning of the productive life. Another difference between these extraordinary does and the average population was their higher fat mobilization between the second AI and the first weaning (this was recovered between weaning and the third AI) [52]. These results partially differed from those of Theilgaard et al. [54], who indicated that there was no positive effect of perirenal fat at the first AI on reproductive life. In fact, a higher risk of culling was associated with high fat mobilization, although does that were too lean also seemed to increase their risk of culling. Similarly, Castellini et al. [50], using perirenal fat, found that does that were too fat and too lean (at AI) showed the poorest fertility. Recent results confirmed the negative effect of fatness at the first AI on the risk of being culled and litter size [14]. The disagreement among these studies and the current one might be related to the different fatness range (probably the absence of does that are too fat at the first AI in our work: maximal fat content at the first AI: 22.1%; Table 2), which might be associated with the time of the first AI (in the latter studies, nulliparous does were inseminated later than in the current one). Body fat and energy at the second AI was also related to fertility in the second AI, which might also reflect the observed positive influence of initial body condition on fertility at the second AI.

Quevedo et al. [55] suggested that the success of AI 11 days after parturition was conditioned by the rabbit doe’s body condition at parturition rather than at insemination. This fact was not observed in this work, probably due to the fact that in our work, we measured body condition at parturition three days before birth (to avoid disturbing the doe) instead of after birth. This prevented the recording of an important proportion of fat mobilization, which was described by Savietto et al. [17]. Once confirmed, the BIA measurement did not alter the doe immediately after parturition (unpublished results); subsequent studies recorded it just after parturition of the doe [38,52].

The reproductive success at the beginning of reproductive life also influenced the fertility of the third AI. In fact, rabbits that are reproductively successful during the first two parturitions were more vulnerable, and consequently, their reproductive performance was impaired in the third AI if they did not have the opportunity to recover. In this sense, Castellini et al. [50,56] proposed the delaying of the second AI after weaning in order to allow rabbit does to recover properly from the first gestation lactation and continue their growth. Otherwise, reproductive success at the beginning of the reproductive life of rabbit does, when they cannot recover their body reserves, might worsen the rabbit doe’s productivity and shorten its life span [13].

The absence of effects of the breeding system and farm may be explained by the synchronization of rabbit does at the moment of AI. Rebollar et al. [57] did not register a rhythm effect on fertility in the second parturition when using controlled lactation as the does’ synchronization tool. However, in experiments without any synchronization method [56,58], it was concluded that the reproductive rhythm was related to fertility rate. Anyway, it must be stressed that the current study was not designed specifically to study the effect of breeding system on fertility rate. Besides, an influence of the breeding systems on the body condition could not be ruled out and further studies would be required to figure out the nature of their relationship.

There were also a positive influence of body protein and energy at the first AI on the percentage of kits born alive at the first parturition. Rommers et al. [15,59] also related a higher protein content at the first insemination, and lower fat content, with a trend towards increasing numbers of kits born alive and percentages of kits born alive at the first parturition. They observed that restricted nulliparous rabbit does inseminated at 17.5 weeks, compared with those fed ad libitum and inseminated at 14.5 weeks, showed higher protein and lower fat content with a similar live weight, and tended to increase the number of kits born alive and the percentage of kits born alive at the first parturition. In primiparous rabbit does, a better body condition (higher body protein, lipid, and energy contents) was also related with changes in metabolic signals (increase in serum protein and leptin concentrations) that might influence ovarian follicle and gamete quality and might be associated with an improved reproductive outcome [60]. Similarly, in sows, ovarian activity and oocyte quality were influenced by the body protein content [61,62]. Another explanation may involve body protein being related to fetal survival, as the increase in litter size has been related to a higher fetal survival, independently of ovulation rate [63].

Live weight and body fat content were also positively correlated with body energy content at the first AI, but negatively with body protein, and had no effect on the percentage of kits born alive. In contrast, Rommers et al. [46] reported that heavier females at the first AI (>4.0 kg) decreased the percentage of kits born alive, although this was combined with an improvement in the litter size at the first parturition. They related it to the development of the reproductive apparatus (larger uterine horns and more *corpora lutea* in the ovaries). Moreover, these results did not agree with those of Quevedo et al. [44], where rabbit does with higher perirenal fat thickness at 3 months of age tended to increase their percentage of kits born alive at the first parturition.

The reduction in the percentage of kits born alive in the second parturition, and the increase in the number of kits born alive, agreed with the results reported by Rommers et al. [46], who also observed, at the second parturition, a higher number of kits born alive and a lower percentage of kits born alive with respect to the first parturition. The different percentage of kits born alive observed in the two farms might be due to the different hybrids used and/or the different environmental management conditions in each farm. No effect of the breeding system or farm on percentage of kits born alive at the second parturition was detected. Weight and body composition at the first AI were not related to this trait. However, higher body protein at the second AI increased the percentage of kits born alive. This result is similar to that recorded at the first AI (on the percentage of kits born alive at the first parturition) and again suggests a positive role of nitrogen content on conception success and fetus viability.

## 5. Conclusions

An adequate body chemical composition at the first AI (around 18% protein and 20% fat) allowed a better fertility at the second AI. However, the consecutive reproductive success at the first and second AI did not allow rabbit does to recover body reserves and impaired fertility at the third AI. Body composition also affected the percentage of kits born alive at the first AI and at the second AI that increased with body protein content.

Consequently, rearing management (e.g., time of first AI) is key to avoiding low fertility rates in primiparous rabbit does. Furthermore, when reproductive success is reached, rabbit does may require alternative management strategies to recover their body condition. Finally, determining body composition at moments of AI may be an adequate tool to anticipate the reproductive success possibilities of rabbit does. Further studies considering different ages at the first AI and breeding systems are warranted to confirm these conclusions and to clarify the relationship between breeding systems and body conditions.

## Figures and Tables

**Table 1 animals-11-02299-t001:** Description of the animals and breeding systems used.

Farm	Rhythm	Number and Strain	Artificial Insemination, d after Parturition	Weaning, d after Parturition
A	R4	37 UPV	4 d	32 d
A/B	R11	28 UPV/31 Hyplus	11 d	35 d
A	R14	41 UPV	14 d	42 d

**Table 2 animals-11-02299-t002:** Estimated chemical body composition ^1^ of nulliparous and primiparous rabbit does during the first three reproductive cycles at artificial insemination and parturition.

	1st Artificial Insemination (N = 137)						1st Parturition (N = 137)					
	LW	Moist.	Prot.	Fat	Ash	Ener.	LW	Moist.	Prot.	Fat	Ash	Ener.
Mean	3906	60.1	17.4	16.1	2.9	1067	4323	60.2	17.3	15.4	2.9	1048
SD ^2^	458	3.2	0.5	2.6	0.15	219	371	3.4	0.6	4.8	0.11	261
Min.	2720	53.4	15.0	8.7	2.7	373	3140	52.4	15.2	2.4	2.7	294
Max.	4836	76.4	18.6	22.1	3.4	1403	5321	77.3	19.0	28.5	3.2	1414
	**2nd Artificial Insemination (N = 132)**						**2nd Parturition (N = 96)**					
	**LW**	**Moist.**	**Prot.**	**Fat**	**Ash**	**Ener.**	**LW**	**Moist.**	**Prot.**	**Fat**	**Ash**	**Ener.**
Mean	4187	60.1	17.4	15.8	3.0	1077	4231	60.2	17.8	15.6	3.1	1125
SD	465	3.6	0.6	3.3	0.12	247	444	3.4	1.0	3.0	0.18	137
Min.	3055	51.6	14.9	6.2	2.7	293	3050	52.4	15.1	3.5	2.5	423
Max.	5280	72.4	19.1	23.1	3.3	1455	5397	77.3	19.8	22.0	3.6	1421
	**3rd Artificial Insemination (N = 96)**						**3rd Parturition (N = 84)**					
	**LW**	**Moist.**	**Prot.**	**Fat**	**Ash**	**Ener.**	**LW**	**Moist.**	**Prot.**	**Fat**	**Ash**	**Ener.**
Mean	4221	59.2	18.1	16.1	3.1	1159	4357	61.1	17.7	15.0	3.1	1091
SD	481	2.9	0.8	2.7	0.15	118	456	3.7	0.9	3.5	0.15	155
Min.	3219	49.8	15.7	8.4	2.6	848	3054	53.8	14.8	5.1	2.7	662
Max.	5680	66.8	19.5	26.0	3.4	1574	5126	71.6	19.7	22.5	3.6	1363

^1^ LW: Live body weight (g); Moist. (moisture); Prot. (protein); fat and ash: % LW; Ener. (body energy): kJ/100 g body weight. ^2^ SD: Standard deviation; Min.: Minimal; Max.: Maximal.

**Table 3 animals-11-02299-t003:** Correlation among estimated chemical body constituents ^1^ during the first three reproductive cycles at insemination and parturition ^2^.

	1st Artificial Insemination, N = 137					1st Parturition, N = 137					2nd Artificial Insemination, N = 132				
	LW	Prot.	Fat	Moist.	Ash	LW	Prot.	Fat	Moist.	Ash	LW	Prot.	Fat	Moist.	Ash
Prot.	−0.46 ***	1				NS	1				NS	1			
Fat	0.83 ***	−0.21 *	1			0.75 ***	0.30 ***	1			0.75 ***	0.25 **	1		
Moist.	−0.59 ***	−0.28 **	−0.95 ***	1		−0.59 ***	−0.57 ***	−0.97 ***	1		−0.72 ***	−0.35 ***	−0.99 ***	1	
Ash	−0.89 ***	0.64 ***	−0.66 ***	0.37 ***	1	NS	0.30 ***	NS	NS	1	−0.78 ***	0.45 ***	−0.62 ***	0.51 ***	1
Ener.	0.65 ***	NS	0.79 ***	−0.78 ***	−0.45 ***	0.63 ***	0.38 ***	0.83 ***	−0.85 ***	NS	0.71 ***	0.36 ***	0.85 ***	−0.89 ***	0.37 ***
	**2nd Parturition,** **N = 96**					**3rd Artificial Insemination,** **N = 96**					**3rd Parturition, N = 84**				
	**LW**	**Prot.**	**Fat**	**Moist.**	**Ash**	**LW**	**Prot.**	**Fat**	**Moist.**	**Ash**	**LW**	**Prot.**	**Fat**	**Moist.**	**Ash**
Protein	−0.62 ***	1				−0.52 ***	1				−0.59 ***	1			
Fat	0.58 ***	NS	1			0.54 ***	NS	1			0.43 ***	NS	1		
Moisture	−0.38 ***	−0.27 **	−0.93 ***	1		−0.42 ***	NS	−0.97 ***	1		−0.35 **	NS	−0.97 ***	1	
Ash	−0.83 ***	0.83 ***	−0.51 ***	0.19 ^†^	1	−0.82 ***	0.80 ***	−0.56 ***	0.35 **	1	−0.77 ***	0.74 ***	−0.57 ***	0.40 ***	1
Ener.	0.48 ***	NS	0.96 ***	−0.98 ***	−0.3 **	0.51 ***	NS	0.98 ***	−0.98 ***	−0.47 **	0.43 ***	NS	0.98 ***	−0.99 ***	−0.48 ***

^1.^ LW: Live body weight (g); Moist. (moisture); Prot. (protein); fat and ash are expressed in % LW; Ener. (energy): KJ/100 g body weight. ^2^ ***: *p* < 0.001; **: *p* < 0.01; *: *p* < 0.05; ^†^: 0.05 < *p* < 0.10.

**Table 4 animals-11-02299-t004:** Correlation among estimated protein, fat and energy body contents at artificial insemination (AI) and parturition during the first three cycles ^1^.

	Protein					Fat					Energy				
	1st AI	1st Partum	2nd AI	2nd Partum	3rd AI	1st AI	1st Partum	2nd AI	2nd Partum	3rd AI	1st AI	1st Partum	2nd AI	2nd Partum	3rd AI
1st partum	NS ^2^	1				0.57 ***	1				0.82 ***	1			
2nd AI	NS	0.38 ***	1			0.33 ***	0.69 ***	1			0.74 ***	0.89 ***	1		
2nd partum	NS	−0.28 **	NS	1		0.37 **	0.48 ***	0.28 **	1		0.29 **	0.44 ***	0.23 *	1	
3rd AI	NS	−0.21 *	NS	0.73 ***	1	0.24 **	0.18 ^†^	0.24 *	0.37 ***	1	NS	NS	0.21 *	0.35 **	1
3rd partum	NS	NS	NS	NS	NS	NS	0.32 **	NS	0.39 **	0.26 **	0.36 **	0.32 **	NS	0.31 **	0.26 **

^1^ N = 137, 132, 96, 96 and 84 rabbit does for 1st partum, 2nd AI, 2nd partum, 3rd AI and 3rd partum, respectively. ^2^ NS: Not significant (*p* > 0.05). ***: *p* < 0.001; **: *p* < 0.01; *: *p* < 0.05; ^†^: 0.05 < *p* < 0.10.

**Table 5 animals-11-02299-t005:** Estimated chemical body composition of nulliparous pregnant and non-pregnant rabbit does at the first artificial insemination (AI) ^1^.

Parameter		Pregnant	Non-Pregnant	RMSE	*p*-Value
Live weight, kg	AI	3.81	3.70	0.43	0.44
	Parturition ^2^	4.31	4.26	0.37	0.74
	Parturition–AI	0.49	0.56	0.29	0.47
Protein%	AI	17.5	17.7	0.48	0.29
	Parturition	17.3	18.8	0.60	<0.001
	Parturition–AI	−0.27	1.08	0.76	<0.001
Fat%	AI	15.6	15.7	2.91	0.95
	Parturition	15.7	19.4	3.00	<0.001
	Parturition–AI	0.15	3.78	2.48	<0.001
Moisture%	AI	60.8	60.3	4.41	0.71
	Parturition	61.2	55.7	4.07	<0.001
	Parturition–AI	0.35	−4.62	4.98	0.003
Ash%	AI	3.01	3.05	0.15	0.34
	Parturition	2.97	3.08	0.11	0.003
	Parturition–AI	−0.041	0.021	0.14	0.18
Energy, kJ/100 g	AI	1067.6	1161.4	214.2	0.19
	Parturition	1062.5	1336.4	249.8	0.001
	Parturition–AI	−0.77	177.0	136.3	<0.001

^1^ N = 128 and 9 for pregnant and non-pregnant does, respectively. ^2^ Measured 3 d before parturition. RMSE: root mean square error.

**Table 6 animals-11-02299-t006:** Effect of estimated body composition at the first insemination on fertility in the second insemination ^1^.

Constituent	Levels	Fertility%	SE ^2^	LL ^3^	UL ^3^	*p*-Value	Q_L_ ^4^
Protein%						0.007	1.33
	16	19.9	0.09	9.1	45.7		
	17	44.5	0.06	33.2	56.2		
	18	72.0	0.06	57.2	83.0		
Fat%						0.030	1.36
	10	31.3	0.10	14.7	54.5		
	15	50.4	0.05	40.6	60.3		
	20	69.5	0.07	54.0	81.4		
Energy, kJ/100 g						<0.001	1.26
	700	19.1	0.07	8.5	37.5		
	1000	46.8	0.05	36.8	56.9		
	1400	83.6	0.06	69.0	92.0		
g protein/MJ energy						<0.001	1.27
	12	78.1	0.06	64.3	87.5		
	17	54.0	0.05	44.5	62.7		
	30	6.10	0.05	1.2	26.1		

^1^ N = 132. ^2^ SE: Standard error. ^3^ LL and UL: lower and upper limits for a confidence interval (95%). ^4^ Q_L_: Deviance of the models.

**Table 7 animals-11-02299-t007:** Effect of estimated body composition at the second AI on fertility in the second artificial insemination ^1^.

Constituent	Level	Fertility%	SE ^2^	LL ^3^	UL ^3^	*p*-Value	Q_L_ ^4^
Fat%						<0.001	1.26
	10	25.2	0.08	13.1	42.8		
	15	53.7	0.05	44.3	62.7		
	20	80.0	0.05	66.8	88.8		
Energy, kJ/100 g						<0.001	1.22
	800	29.3	0.07	17.4	45.0		
	1100	58.4	0.05	73.7	67.0		
	1300	76.1	0.05	64.8	84.4		
g protein/MJ energy						<0.001	1.23
	12	75.1	0.05	63.6	83.7		
	17	58.9	0.05	49.7	67.5		
	30	17.0	0.08	6.4	38.2		

^1^ N = 132. ^2^ SE: Standard error. ^3^ LL and UL: lower and upper limits for a confidence interval (95%). ^4^ Q_L_: Deviance of the models.

**Table 8 animals-11-02299-t008:** Effect of live weight at the first artificial insemination (AI) and its increment from the first insemination to the first parturition (model 1) or from the first to the second inseminations (model 2) on fertility at the second parturition.

	Model 1 (N = 137. QL1 = 1.33) ^1^			Model 2 (N = 132. QL1 = 1.29)		
Weight at 1st AI, g	Increment-I2 ^2^	Fertility%	SE	Increment-II3 ^3^	Fertility%	SE ^4^
3000	−2	9.98	0.08	−5	14.6	0.09
	10	21.7	0.10	15	35.2	0.10
	20	37.2	0.10	30	56.4	0.10
3500	−2	19.4	0.09	−5	25.7	0.08
	10	37.6	0.08	15	52.3	0.07
	20	56.4	0.07	30	72.3	0.09
4000	−2	34.4	0.08	−5	41.2	0.07
	10	56.7	0.05	15	68.9	0.05
	20	73.8	0.07	30	84	0.06
4500	−2	53.4	0.09	−5	58.5	0.08
	10	74.1	0.08	15	81.7	0.07
	20	85.9	0.07	30	91.4	0.05

^1^ Q_L_: Deviance of the models. ^2^ Increment-I: increment of body weight between the first AI and the first parturition expressed as a percentage of body weight at the first AI. ^3^ Increment-II: increment of body weight between the first and second AI expressed as a percentage of body weight at the first AI. ^4^ SE: standard error. Model 1: *p*-_weight_ = 0.010; *p*-_increment1_ = 0.006. Model 2: *p*-_weight_ = 0.010; *p*-_increment2_ = 0.001.

**Table 9 animals-11-02299-t009:** Effect of estimated body protein content at the third artificial insemination (AI), its increments and fat increment on fertility at the third artificial insemination ^1^.

Constituent	Level	Fertility%	SE ^2^	LL ^3^	UL ^3^	*p*-Value	Q_L_ ^4^
Protein ^5^%						0.030	1.11
	16	38.9	0.10	13.6	72.0		
	17	56.6	0.10	36.8	74.4		
	18	72.5	0.04	62.2	80.8		
Protein gain (AI3–AI1) ^5^						0.021	1.09
	−5	56.8	0.10	39.3	72.9		
	5	69.1	0.05	58.1	78.4		
	15	79.2	0.07	68.0	87.2		
Protein gain (AI3–P2) ^6^						<0.001	0.99
	−5	57.8	0.07	43.4	37.5		
	5	86.5	0.05	74.9	93.2		
	15	96.7	0.06	86.7	99.2		
Fat gain (AI3–P2) ^7^						0.020	1.08
	−20	61.5	0.06	46.4	74.7		
	20	81.2	0.05	68.9	89.4		
	50	90.1	0.05	72.9	96.8		

^1^ N = 96. ^2^ SE: standard error. ^3^ LL and UL: lower and upper limits for a confidence interval (95%). ^4^ Protein content at 3rd AI. ^5^ Protein increment between the moments of the first and third AI expressed as a percentage of the initial protein content.^6^ Protein increment between the moments of third AI and second parturition expressed as a percentage of protein content at the second parturition. ^7^ Fat increment between the moments of the third AI and second parturition expressed as a percentage of fat content at the second parturition.

**Table 10 animals-11-02299-t010:** Effect of the change of live body weight and body composition, between the second parturition and the third artificial insemination (AI), and fertility at the second AI on the reproductive success at the third artificial insemination ^1^.

Change between 2nd Parturition and 3rd AI2 ^2^	Pregnant at Third AI	Non-Pregnant at Third AI	RMSE ^3^	*p*-Value
Live body weight	2.4	−4.7	8.08	<0.001
Protein	3.2	−3.3	7.08	<0.001
Fat	42.3	−5.1	210	NS ^4^
Energy	6.8	−4.8	39.7	NS
Fertility at the second AI	62.0	84.0	-	0.043

^1^ N = 96. ^2^ expressed as a percentage of the content at 2nd parturition. ^3^ RMSE: root mean square error. ^4^ NS: non-significant.

**Table 11 animals-11-02299-t011:** Effect of body protein and energy content at the first insemination on the percentage of kits born alive at the first parturition ^1^.

Constituent	Level	Kits Born Alive%	SE ^2^	LL ^3^	UL ^3^	*p*-Value	Q_L_ ^4^
Protein%						0.040	2.94
	16	88.7	0.03	0.82	0.95		
	17	92.5	0.01	0.91	0.96		
	18	95.2	0.01	0.94	0.97		
Energy, kJ/100 g						0.010	2.94
	900	92.4	0.01	0.91	0.96		
	1100	93.8	0.01	0.93	0.98		
	1300	95.0	0.01	0.94	0.97		

^1^ N = 121. ^2^ SE: standard error. ^3^ LL and UL: lower and upper limits for a confidence interval (95%). ^4^ Deviance of each model.

**Table 12 animals-11-02299-t012:** Effect of body protein content at the second insemination on percentage of kits born alive at the second parturition ^1^.

Constituent	Levels	Kits Born Alive%	SE ^2^	LL ^3^	UL ^3^	*p*-Value	Q_L_ ^4^
Protein%						<0.001	5.12
	16	71.0	0.06	57.0	81.7		
	17	83.3	0.02	79.0	86.6		
	18	91.0	0.01	88.3	93.1		

^1^ N = 83. ^2^ SE: standard error. ^3^ LL and UL: lower and upper limits for a confidence interval (95%). ^4^ Deviance of the model.

## Data Availability

The data presented in this study are available on request from the corresponding author. The data are not publicly available due to confidentiality requirements agreed with one of the farms.

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
