# Peer review of "Relationship between Body Chemical Composition and Reproductive Traits in Rabbit Does"

_animals, 2021, doi:10.3390/ani11082299_

Round 1

Reviewer 1 Report

In this article, the authors examined the effects the factors related to body chemical composition on fertility and offspring survival at birth to identify the threshold to be required for female rabbits at the beginning of their reproductive life. They demonstrated rearing management, such as time of first AI, is important to avoid low fertility rates in primiparous rabbit. They also indicated the usefulness of body composition analysis at AI for reproductive success on rabbit.

The individual factors of body chemical composition on rabbit fertility are well discussed and the results are interesting, but I think there are some minor revisions that should be made before publication.

  1. Results. I think it is better to rewrite more conciseness, but not required.
  2. Table 3 & 4. Please uniform the display of the correlation among same factor (1). Showing all or nothing.
  3. Line 224. Please correct the word “AD” into “AI”.
  4. Line 229. Please correct the number “17” into “16”.
  5. Line 281. Please correct the word “QL1” into “1QL”.
  6. Line 285. I don’t know whether the insertion position of the sentence is right.
  7. Line 411. I recommend to rewrite the term “similar than” into “similar to”.
  8. Line 450. I recommend to insert the term “it was” before “concluded”.
  9. Line 457. I recommend to rewrite the term “respect to” into “, compared with”.

Author Response

We appreciate the suggestions made by the reviewer.

Results. I think it is better to rewrite more conciseness, but not required.

We’d rather prefer to avoid any reduction in order to facilitate the understanding of the discussion.

Table 3 & 4. Please uniform the display of the correlation among same factor (1). Showing all or nothing.

It was corrected.

Line 224. Please correct the word “AD” into “AI”.

It was corrected.

Line 229. Please correct the number “17” into “16”.

It was corrected.

Line 281. Please correct the word “QL1” into “1QL”.

It was corrected.

Line 285. I don’t know whether the insertion position of the sentence is right.

It was corrected the insertion position.

Line 411. I recommend to rewrite the term “similar than” into “similar to”.

It was corrected.

Line 450. I recommend to insert the term “it was” before “concluded”.

It was corrected.

Line 457. I recommend to rewrite the term “respect to” into “, compared with”.

It was corrected.

Reviewer 2 Report

The authors performed interesting research. Their task was to establish the relationship between the chemical composition of the body (during insemination and childbirth), determined by bioelectric impedance, both with fertility and the percentage of young born alive during the first three inseminations, and the determination of the most important moments for insemination. 

When reading the study, many questions arise: - what was the rabbit feeding system, - what were their components (%), the physico-chemical composition of the feed, energy, or nutrition was balanced in accordance with the nutrition standards - which is a significant impact on the health of rabbits, their condition Animal diet is a valuable source of ingredients affecting health, often probiotic supplements play a preventive role in various disorders. The authors do not provide such information. - what was the sanitary condition of the feed provided, the presence of mycotoxins. Similarly, in terms of the microclimate, little information has been provided, please elaborate, maintenance conditions, presence of gaseous pollutants, ventilation? It is also puzzling that the research was conducted on the basis of the principles of animal care in experiments - Spanish Royal Decree 1201/2005 - in 2010, i.e. 11 years ago? on two different farms, and the conditions were different, and the nutrition? 

please supplement, clarify 

Author Response

We appreciate the suggestions made by the reviewer.

The feeding management was similar in the two farms although they used different commercial diets (now they are described in the manuscript).

When reading the study, many questions arise:

- what was the rabbit feeding system:

It was indicated that rabbit does were fed ad libitum, but those not lactating-not pregnant were restricted. In this way, we corrected a mistake because rabbit does were all fed ad libitum after the first insemination.

- what were their components (%), the physico-chemical composition of the feed, energy, or nutrition was balanced in accordance with the nutrition standards - which is a significant impact on the health of rabbits, their condition Animal diet is a valuable source of ingredients affecting health, often probiotic supplements play a preventive role in various disorders:

The main ingredients and chemical composition are now included. 

 The authors do not provide such information. - what was the sanitary condition of the feed provided, the presence of mycotoxins.

This was not measured. In our experience, even when there are suspicions of mycotoxins contamination it is difficult to record it.

Similarly, in terms of the microclimate, little information has been provided, please elaborate, maintenance conditions, presence of gaseous pollutants, ventilation?

Farms had forced ventilation, and heating/cooling systems and it is now included in the manuscript.

No measurement was made of gaseous pollutants in the farms.

It is also puzzling that the research was conducted on the basis of the principles of animal care in experiments - Spanish Royal Decree 1201/2005 - in 2010, i.e. 11 years ago?

The principles of animal care standards followed in this experiment were exactly the same than those employed in more recent studies that received the approval of the homologated ethics committee according to the latest UE legislation (Delgado et al., 2017; Referenced in the manuscript as: [52]; or Delgado et al., 2018. J. Anim. Sci. 96:1084–1100).

Reviewer 3 Report

Dear Authors,

I congratulate you on your original study which uses not only an innovative tool to determine the body condition of rabbits, but also a refined statistical approach. I also find that evaluating the associations between successive inseminations is very appropriate.

I have no detailed corrections to suggest but just some food for thought that I would like to discuss with you.

My biggest doubt is the use of different rhythms of AI. I imagine that you have chosen different breeding systems to increase the variability of body conditions (if so, it could be specified on Materials and methods section). Despite the synchronization of estrus and the fact that the inclusion in the statistical model of the breeding system should "hold constant" this factor, I fear that it could still represent a confounding factor. Moreover, I suppose that there may be a significant association between the breeding system and body conditions so that the independence between these two factors included as independent variables could lack. Please check this association before including both as independent variables. This could indeed create a bias in the results and “mask” the effect of the breeding system. Have you thought to stratify the analyzes according to the insemination rhythm? This would reduce the variability and the range of validity of each model, but would eliminate the influence of a potential confounding factor. Always about the rhythm, have you evaluated the receptivity? Were prostaglandins included in your synchronization protocol? According to you, could it be interesting to evaluate progesterone levels to rule out interference from high progesterone (P+) syndrome? For example, Theau Clément et al. showed that 4d after parturition (your R4) does are characterised by a low receptivity and a high level of peripheral plasma progesterone (Theau Clément et al. 2000 Description of the ovarian status and fertilising ability of primiparous rabbit does at different lactation stages. WORLD RABBIT SCIENCE 8 supp. 1, pp 259-266). In conclusion, I do not know if your experimental protocol does not allow to separate the effects of body conditions and other factors associated with the day of AI. Thus, although I agree on the most probable effect of body conditions, an influence of the breeding systems could not be ruled out as the lack of significance that you have found could be due to the association between breeding systems and body conditions (as they can share some of the variance).

I also report:

- repetition of the results about protein content at the first AI to the lines 229-230 and 235-236.

- please report the fertility obtained with the 3 reproductive rhythms for each cycle of AI

Moreover, I suggest to include a short paragraph on the limitations of the study and ideas for future studies (for example: to apply the present protocol (bioelectrical impedance and evaluation of successive inseminations) on rabbits of very different ages and compositions at the first AI to establish the body composition that allows the best future performance. For nulliparous does you would not have the problem of the postpartum day as a confounding factor).

My final suggestion is, in general, to try to simplify the Results and lighten the Discussions. I understand that the models are complicated, but these sections are a bit tricky and not easy to read.

Author Response

Thank you for your very proper comments and suggestions. We answer you below:

My biggest doubt is the use of different rhythms of AI. I imagine that you have chosen different breeding systems to increase the variability of body conditions (if so, it could be specified on Materials and methods section).

We have now included it.

Despite the synchronization of estrus and the fact that the inclusion in the statistical model of the breeding system should "hold constant" this factor, I fear that it could still represent a confounding factor.

We agree with the reviewer, and the text already contained the sentence: ‘Anyway, it must be stressed that the current study was not designed specifically to study the effect of breeding system on fertility rate’.

Moreover, I suppose that there may be a significant association between the breeding system and body conditions so that the independence between these two factors included as independent variables could lack. Please check this association before including both as independent variables.

We agree with the reviewer, and we would have expected that relationship, but we did not clearly find it (P ³ 0.14), and for that reason we avoided to include that approach. Several considerations regarding this comment: i) in the first AI (where most rabbit does get pregnant) the breeding system had no influence, ii) in the second and third AI the fertility was lower (and they was an effect of the breeding system in the third one -now indicated in the text-). Differences in fertility in the second AI possibly produced higher differences in body condition than those derived directly from the breeding system, iii) the model could have been more complex including the effect of time, but the repeated measurements model was difficult to manage (considering the differences in the times among the breeding systems). For these reasons we tried to be especially cautious with the results derived from the second and third AI.

This could indeed create a bias in the results and “mask” the effect of the breeding system. Have you thought to stratify the analyzes according to the insemination rhythm? This would reduce the variability and the range of validity of each model, but would eliminate the influence of a potential confounding factor.

Yes. However, and considering the previous comments (and p-values) we refused to do it. The conclusions regarding the body condition at first AI are reasonably clear, and those regarding the second and third AI are not because they are affected by more factors and probably would require a larger number of replicates (and combined with an even more complex model).

Always about the rhythm, have you evaluated the receptivity?

No, we didn’t.

Were prostaglandins included in your synchronization protocol?

No, prostaglandins were not included in our synchronization protocol. It was also included in the text: “Seminal doses with more than 20 million spermatozoa in 0.5 ml of a commercial diluent (Magapor S.L.) were made using a pool of fresh heterospermic semen from bucks selected for growth performance”.

According to you, could it be interesting to evaluate progesterone levels to rule out interference from high progesterone (P+) syndrome? For example, Theau Clément et al. showed that 4d after parturition (your R4) does are characterised by a low receptivity and a high level of peripheral plasma progesterone (Theau Clément et al. 2000 Description of the ovarian status and fertilising ability of primiparous rabbit does at different lactation stages. WORLD RABBIT SCIENCE 8 supp. 1, pp 259-266).

We agree with you, but we have not measured progesterone level in this work.

 In conclusion, I do not know if your experimental protocol does not allow to separate the effects of body conditions and other factors associated with the day of AI. Thus, although I agree on the most probable effect of body conditions, an influence of the breeding systems could not be ruled out as the lack of significance that you have found could be due to the association between breeding systems and body conditions (as they can share some of the variance).

Yes, we agree, and we have included the sentence: ‘Besides, an influence of the breeding systems on the body condition could not be ruled out and further studies would be required to figure out the nature of their relationship’.

I also report:

- repetition of the results about protein content at the first AI to the lines 229-230 and 235-236.

It was corrected.

- please report the fertility obtained with the 3 reproductive rhythms for each cycle of AI

It is now included the fertility in the third AI.

Moreover, I suggest to include a short paragraph on the limitations of the study and ideas for future studies (for example: to apply the present protocol (bioelectrical impedance and evaluation of successive inseminations) on rabbits of very different ages and compositions at the first AI to establish the body composition that allows the best future performance. For nulliparous does you would not have the problem of the postpartum day as a confounding factor).

We included in the conclusions the sentence: ‘Further studies considering different ages at first AI and breeding systems are warranted to confirm these conclusions and to clarify the relationship between breeding system and body condition’.

My final suggestion is, in general, to try to simplify the Results and lighten the Discussions. I understand that the models are complicated, but these sections are a bit tricky and not easy to read.

We are open to simplify the manuscript, but we do not know exactly what part, because it would imply to remove some tables.

Round 2

Reviewer 3 Report

Dear Authors,

thank you for your answer. I believe the paper can now be published. I still have doubts about the confounding effect of the insemination rhythm, but I it can be a starting point for future studies.

Author Response

As we have already commented, we agree with the reviewer. The use of different rhythms was used to obtain wider differences in body chemical composition and fertility values, but this trial was not specifically designed to study this relationship (for that purpose the three rhythms should have been represented in the two farms). For this reason, the main conclusion of this trial is focused on the body chemical composition at first insemination (because later there is a more complex scenario with more variables implicated). We indicate that further studies are required, although to study that relationship more complex statistical models will be necessary (and probably a higher number of replicates).